# Covalently-assembled single-chain protein nanostructures with ultra-high stability

Wenqin Bai[1], Cameron J. Sargent [2], Jeong-Mo Choi [3], Rohit V. Pappu [3] & Fuzhong Zhang [1,2,4]

Protein nanostructures with precisely defined geometries have many potential applications in catalysis, sensing, signal processing, and drug delivery. While many de novo protein nanostructures have been assembled via non-covalent intramolecular and intermolecular interactions, a largely unexplored strategy is to construct nanostructures by covalently linking multiple individually folded proteins through site-specific ligations. Here, we report the synthesis of single-chain protein nanostructures with triangular and square shapes made using multiple copies of a three-helix bundle protein and split intein chemistry. Coarse-grained simulations confirm the experimentally observed flexibility of these nanostructures, which is optimized to produce triangular structures with high regularity. These single-chain nanostructures also display ultra-high thermostability, resist denaturation by chaotropes and organic solvents, and have applicability as scaffolds for assembling materials with nanometer resolution. Our results show that site-specific covalent ligation can be used to assemble individually folded proteins into single-chain nanostructures with bespoke architectures and high stabilities.

[1] Department of Energy, Environmental and Chemical Engineering, Washington University in St. Louis, Saint Louis, MO 63130, USA. [2] Division of Biological & Biomedical Sciences, Washington University in St. Louis, Saint Louis, MO 63130, USA. [3] Department of Biomedical Engineering and Center for the Science & Engineering of Living Systems, Washington University in St. Louis, Saint Louis, MO 63130, USA. [4] Institute of Materials Science & Engineering, Washington University in St. Louis, Saint Louis, MO 63130, USA. Correspondence and requests for materials should be addressed to F.Z. (email: fzhang@seas.wustl.edu)

Over the past decade, the design rules and computational tools for DNA origami have been well established, and a wide variety of intricate and impressive DNA-based nanostructures have been demonstrated[1–4]. In recent years, much attention has also been turned towards the creation of complex and tailor-made nanostructures using proteins, the class of bio-molecules that nature employs to make both functional and structural materials. Proteins contain a wide variety of chemical functional groups, are capable of complex folding possibilities[5], and can be sustainably produced in vivo. Accordingly, protein-based nanostructures have great potential to be used in valuable applications ranging from drug delivery vehicles[5–7] to scaffolds for assembling inorganic nanoparticles or enzymes for sensing, catalysis, or signal transduction[6,7].

Although the field is still quite nascent, several design strategies currently exist for constructing protein nanostructures. Top-down approaches rely on naturally occurring protein structures, such as bacterial microcompartments, engineering these structures to allow regiospecific labeling of functional molecules and particles or to alter their dimensions and shapes[8]. However, because the extent of changes that can be made to these natural structures is fairly limited, rationally designing tailor-made structures through top-down approaches remains difficult. Meanwhile, bottom-up approaches to create de novo nanostructures using computationally designed protein structures and/or assembly interfaces have greatly expanded the diversity and potential applications of protein structures[5,9]. Despite the difficulty of de novo protein design, recent developments have resulted in the creation of impressive protein-based nanostructures with precisely defined symmetries, geometries ranging from 2D arrays[10] to 3D tetrahedra[5,7,11] and icosahedra[9,12], and length scales ranging from nanometers[5] to hundreds of nanometers[6]. A variety of protein interfaces using non-covalent interactions, such as hydrogen bonding, electrostatic interactions, the hydrophobic effect, and metal-coordinated interactions[5,7,9,10,12–14], have been employed to assemble these complexes. These efforts demonstrate the viability of creating protein-based nanostructures using bottom-up approaches.

A largely unexplored strategy is to link individually folded proteins via site-specific covalent ligation to generate homogeneous and controllable multi-subunit nanostructures. Although covalent ligation has been used in some instances to reinforce protein structures assembled via noncovalent interactions[6,15,16] or to attach other functional proteins to such structures[7,17], the post-translational assembly of proteinaceous nanostructures exclusively through covalent interactions has not been widely explored. Site-specific covalent assembly could result in fewer defects from incorrectly assembled or kinetically trapped intermediate structures, which are often seen in complexes constructed via non-covalent interactions[18]. Additionally, site-specific covalent assembly could be used in conjunction with both top-down and bottom-up approaches, positioning it as a versatile tool to use in creating protein nanostructures. Furthermore, it stands to reason that structures assembled using covalent bonds will generally have higher thermodynamic stability at the assembly interfaces than structures assembled through non-covalent interactions, thus enabling broader applications in extreme temperatures and chemical conditions[19,20].

In this report, we explore this design strategy by covalently connecting individually folded protein blocks end-on-end to form cyclized, single-chain nanostructures with defined geometry. Specifically, we use split inteins (SI) to post-translationally ligate multiple copies of a hyper-stable three-helix bundle (3HB), creating 2D nanostructures with triangle and square shapes. By optimizing the lengths of the peptide linkers between each 3HB copy, we then produce nanostructures of increased uniformity and tunable flexibility. These structures exhibit high stability in harsh denaturing conditions as well as utility as a scaffold for precisely assembling inorganic nanoparticles, demonstrating the merit of covalent assembly in developing protein-based nanostructures with exceptional and customizable properties.

## Results

**Design and synthesis of protein nanostructures.** To construct single-polypeptide chain nanostructures with defined geometry, we used SI chemistry[21] to covalently ligate multiple individually folded protein structures (referred to here as building blocks) into one cyclized structure. Of the various protein ligation approaches, SIs are well-suited for the assembly of protein nanostructures because they are small, autocatalytic domains and have several orthogonal pairings that allow for tightly controlled assembly through sequence-specific interactions. Compatible SIs fused to the termini of two protein building blocks can be used to form a peptide bond between them, leaving only a few scar residues at the ligation site. If building blocks are flanked by orthogonal SIs in the form of $Int^C$-building block-$Int^N$ (with $Int^C$ and $Int^N$ representing the C- and N-terminal halves, respectively, of SI pairs), multiple building blocks can be covalently assembled. The large number of orthogonal SI pairs from both natural sources[21,22] and engineering efforts[23] allows the covalent assembly of multiple building blocks in a defined order. Together with a selection of high-yield, fast-ligating SIs[22,23], we chose to use a recently engineered 3HB protein[24] as a building block to create simple yet highly stable 2D nanostructures. When folded, this 3HB forms a rigid protein rod that is 12 nm long and exceptionally stable (folding energy > 60 kcal/mol). Furthermore, the two termini of the 3HB are located on opposite ends of the rod-shaped protein, meaning that SI groups can be fused to both termini to build the structure bidirectionally. By cyclically ligating together a set number of orthogonally reactive 3HB blocks, 2D shapes with the same number of sides can be constructed (Fig. 1).

To create these structures, we designed and constructed multiple plasmids each encoding a 3HB block fused to non-complementary SI domains on both termini (Supplementary Tables 1 and 2, Supplementary Datasets 1 and 2). Additionally, a proline–glycine dipeptide was placed between 3HB and each of the SI groups of the fusion constructs to prevent the propagation of α-helical structure from the 3HB to the SIs[3]. Three blocks with complementary SIs were prepared: $Gp^C_{18}$-3HB-$Cfa^N_{14}$, $Cfa^C_{14}$-3HB-$NrdJ^N_{18}$, and $NrdJ^C_{18}$-3HB-$Gp^N_{18}$, where Gp, Cfa, and NrdJ are the orthogonal SI pairs[22,23], the superscripts denote the C- or N- half of each SI pair, and the subscripts represent the total number of residues in the corresponding peptide linker between copies of 3HB in the final structure after ligation. Each fusion protein was individually expressed and purified, and the three were then mixed together to form a covalently linked triangular nanostructure called Tri18 (Supplementary Fig. 1). When imaged by scanning transmission electron microscope (STEM; Fig. 2a), each edge of Tri18 was ~12 nm long, agreeing with the reported length of the original 3HB rod and indicating that the three 3HB copies in the assembled nanostructure folded properly. The vertex angles in each structure varied considerably, with a standard deviation of 17.4° from the mean value of 60°, the value expected for each angle in an equilateral triangle (Fig. 2b). In some irregular shapes, the angles deviated from 60° by as much as 40°. However, the edge lengths remained ~12 nm in all measured shapes, suggesting that these angle variations are caused by the flexible linkers rather than partially unfolded 3HB blocks.

During the synthesis of Tri18, we observed a few faint, low molecular weight bands on SDS–PAGE gels (Supplementary Fig. 1, Supplementary Table 3) after purification with affinity

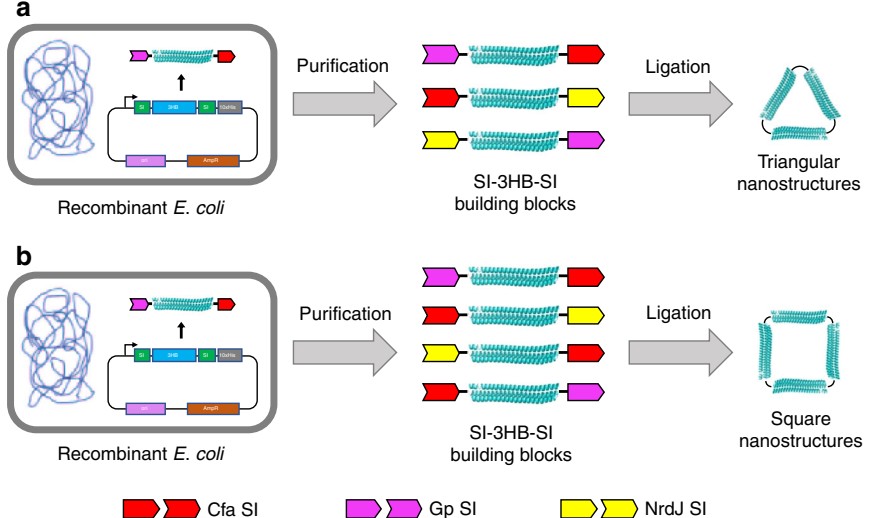

**Fig. 1** Schematic illustration of the covalent assembly of triangle and square protein nanostructures. **a** By genetically fusing cognate SI pairs to the N- and C-termini, three copies of 3HB (PDB:4TQL) can be linked together to form triangular nanostructures. **b** Using constructs with a different combination of the same orthogonal SI pairs, square nanostructures containing four copies of 3HB can also be made

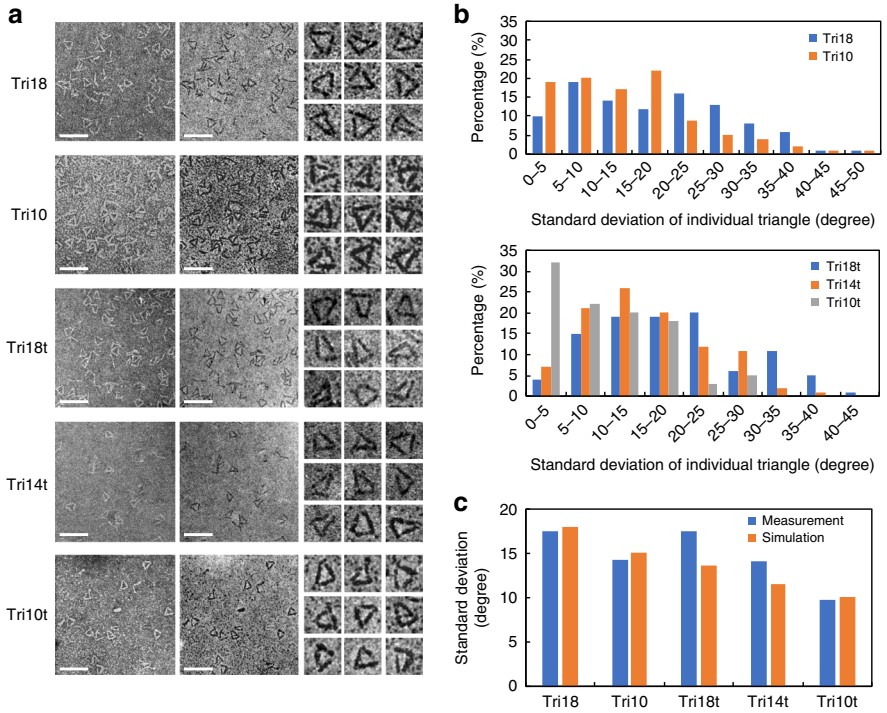

**Fig. 2** Construction of protein triangles and optimization of vertex linker lengths. **a** STEM images of protein triangles. The images in the left column are bright-field overview images of the triangles, the images in the middle column are dark-field overview images of the same triangles, and the images in the right column are magnified dark-field images of individual triangles. Scale bars, 50 nm. Magnified images, 25 nm × 25 nm. **b** Histograms displaying the standard deviations of the three vertex angles within each individual triangle for the five triangular nanostructures made in this study using full length (above) and truncated (below) 3HB constructs ($n = 100$ triangles for each structure). **c** Standard deviations of all individual vertex angles in each of the five constructed triangles from STEM image measurements and computational simulations ($n = 100$ triangles for each structure). Source data are provided as a Source Data file

chromatography and, to a lesser extent, after further purification by size-exclusion chromatography (SEC) (Supplementary Table 3). To identify whether these were unreacted intermediates or side-products, these low molecular weight species were isolated by SEC and further analyzed by SDS–PAGE and STEM

(Supplementary Fig. 2). These species were identified as monomeric, dimeric, and uncyclized trimeric 3HB structures based on their relative SDS–PAGE gel migration rates (i.e. molecular weights; Supplementary Fig. 2b) and STEM images (Supplementary Fig. 2c). The SEC-isolated monomeric 3HB had

an apparent molecular weight ~15 kDa smaller than the unreacted, SI-flanked 3HB building blocks (Supplementary Figs. 1 and 2b), suggesting the loss of both SI domains. This result is in agreement with previous reports of SI side reactions, in which $Int^C$ and/or $Int^N$ groups are cleaved off the proteins prior to normal SI ligation[21]. Cleavage of SI domains without ligation could occur at several stages during the multi-step ligation reaction, generating monomers, dimers, or uncyclized trimers that can no longer react. Meanwhile, we also observed unreacted monomer, which can be explained by previously reported ligation yields of the SIs used (85–95%)[25]. Further engineering of SIs to improve SI ligation yield could improve the overall yield of triangular nanostructures.

Next, to check whether the linkers between 3HB building blocks form any regular secondary structures or instead form heterogeneous, random coil-like ensembles, we performed atomistic simulations for those linkers using the ABSINTH implicit solvation model and forcefield paradigm. This simulation approach has been deployed successfully for uncovering sequence-ensemble relationships of a wide range of intrinsically disordered proteins[26–33]. The results indicate a low level of secondary structure formation in the linkers (<10% of the ensemble in each case is part of an ordered secondary structure; Supplementary Table 4), and that the linkers indeed behave like Gaussian random chains, as their end-to-end distributions can be fitted well to a Gaussian model (Supplementary Fig. 3). End-to-end distance distributions deviate from being Gaussian if the effective solvent quality deviates from the theta point. Concordance with the Gaussian distribution implies that the effective solvent quality is theta-like for the linker sequences. Further, the linker lengths are larger than the Kuhn length but smaller than the thermal blob size, meaning that the chain behavior can therefore be simply explained by ideal-like interactions of Kuhn monomers.

**Fine-tuning the flexibility of triangular nanostructures**. To reduce the flexibility of triangular nanostructures, we truncated the peptides between the 3HB and SI groups to produce linkers in the final structure with 10 residues, yielding $Gp^C_{10}$-3HB-$Cfa^N_{10}$, $Cfa^C_{10}$-3HB-$NrdJ^N_{10}$, and $NrdJ^C_{10}$-3HB-$Gp^N_{10}$ building blocks. Ligation of these three blocks yielded the Tri10 nanostructure (Fig. 2a), which displayed slightly improved regularity (Fig. 2b). The number of residues between the 3HB and SIs were then further reduced to make blocks that would yield a final structure with linkers seven residues long. However, when mixed together, these blocks yielded linear products but no cyclized triangular products, suggesting that the reduced flexibility of these shortened linkers prevented cyclization. Meanwhile, we noticed that, based on the crystal structure, a few residues at the N- and C-termini of 3HB, namely NEDDM (residues 1–5) and GLE (residues 239–241), are not included in the α-helices and therefore do not contribute to the rigid portions of the structure but instead might contribute to the flexibility observed in the linkers. Accordingly, we prepared fusion proteins using a truncated 3HB lacking these residues together with different linker lengths. We ligated these to create three triangular nanostructures, Tri18t, Tri14t, and Tri10t (Supplementary Table 1). STEM imaging showed a dramatic reduction of vertex flexibility in Tri10t, with a large proportion of these structures being close to equilateral triangles (Fig. 2a, b). The uniformity of these triangles demonstrates that our approach can successfully create nanostructures with consistent and reliable dimensions, a necessary characteristic of synthetic nanochemistry.

To understand the relationship between the structural regularity and the linker length, we performed simple coarse-grained simulations in which 3HB was modeled as a rigid rod and the linkers were represented by a two-dimensional Gaussian chain (see "Methods" section). We tested various combinations of spring constants for the linker chains and obtained a linear relationship of the summation of characteristic lengths of three Gaussian-chain linkers ($\Sigma R_0$) to the standard deviation ($\sigma$) of angles (Supplementary Fig. 4):

$$\sigma = -2.37° + \left(0.20°/Å\right) \times \Sigma R_0. \qquad (1)$$

This implies that chain extensibility (as quantified by the characteristic length) almost additively contributes to the triangle flexibility (as quantified by the standard deviation). The characteristic lengths of the experimental linkers can be determined as done above by Gaussian fitting of atomistic simulation data (Supplementary Table 5), which successfully reproduces the experimentally determined standard deviations (Fig. 2c). This shows that the flexibility of a triangular structure can be explained mostly by the near-ideal behavior of linkers, which have the requisite sequence features to ensure this behavior.

**Synthesis of square-shaped nanostructures**. Using the shortest linkers that allowed the formation of triangular structures, we designed four fusion proteins ($Gp^C_{10}$-$3HB_t$-$Cfa^N_{10}$, $Cfa^C_{10}$-$3HB_t$-$NrdJ^N_{10}$, $NrdJ^C_{10}$-$3HB_t$-$Cfa^N_{10}$, and $Cfa^C_{10}$-$3HB_t$-$Gp^N_{10}$) to create square-shaped protein nanostructures (Fig. 3a, Supplementary Figs. 5 and 6). While the distribution of the observed angles in these ligated squares peaked around the mean value of 90°, peaks of nearly the same magnitude around 70° and 110° and a standard deviation of 27.6° were also observed, indicating considerably greater angular freedom in the squares than in the triangles (Fig. 3b). Coarse-grained simulations suggest that simulated squares would show a significantly higher standard deviation (50.7°) than the measured value (27.6°) without any constraint on individual angles. These simulations also predict two asymmetric peaks around 30° and 135° (Supplementary Fig. 7), which can be explained by fluctuations around the rhombus-like nature of many of the structures imaged using STEM. Since the length of 3HB is much longer than the length scale of the linkers, the structure behaves like a rhombus with four equal sides and four angles: $\theta$, $180°-\theta$, $\theta$, and $180°-\theta$, where $\theta$ is a free parameter. Therefore, the linker flexibility not only broadens peaks but also produces a bimodal angular distribution unless it peaks around 90°. The experimentally observed trimodality thus implies that there may be two populations, one close to a perfect square and another with two pairs of angles ~70° and ~110°. The simulations are based on a coarse-grained model and the assumption of rigid rods connected by flexible linkers. While this model captures the overall features of the structural ensemble, the distinct populations inferred from the experimental data, specifically those that yield the apparent trimodality in the $\theta$ parameter distribution, were not observed in the simulated ensemble. It would appear that the discrete states observed experimentally are reflective of the bendability of individual angles that are feasible for the current architecture. This requires further investigation and systematic titrations of sequence parameters of flexible linkers including the amino acid composition, which tunes the persistence length, and the overall lengths of linkers. At this juncture, such titrations are challenging because they directly impact the efficiency of cyclic ligation. Other methods for titrating the sequence parameters of linkers are necessary to probe for ways to increase the uniformity of square structures.

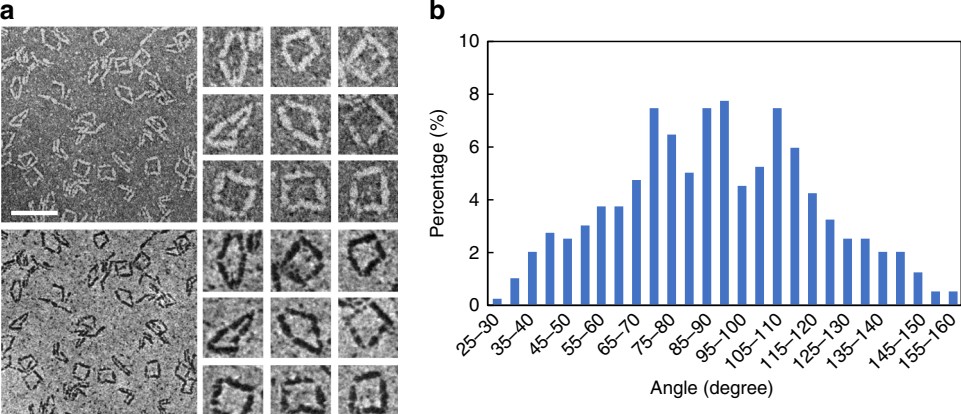

**Fig. 3** Construction of protein squares and their angle distributions. **a** Bright-field (top) and dark-field (bottom) STEM images of the protein squares, with individual squares magnified on the right. Scale bars, 50 nm. Magnified images, 25 nm × 25 nm. **b** Experimental distribution of angles measured in the square nanostructures ($n = 400$, with four angles measured from each of 100 squares). Source data are provided as a Source Data file

**Protein nanostructures display ultra-high thermostability**. We next examined the stabilities of our single-chain nanostructures using the rigid Tri10t. Secondary structure stability was first assessed using circular dichroism (CD). Tri10t in phosphate-buffered saline (PBS; pH 7.4) with an additional 500 mM NaCl was incubated at 90 °C for 4 h, during which little to no unfolding was observed, consistent with the CD spectral changes reported previously for 3HB[24] (Fig. 4a). When the temperature was returned to 20 °C, the CD spectrum fully recovered its original shape, suggesting that there is either no unfolding or no irreversible unfolding of the α-helices in Tri10t at 90 °C over an extended period of time. We then examined the stability of Tri10t at high temperatures using STEM. Tri10t in PBS buffer (pH 7.4) with 500 mM NaCl was incubated at elevated temperatures for 1 h and then immediately prepared for STEM imaging. A large proportion of intact triangles at all tested temperatures and minimal degradation below 80 °C were observed, indicating that the cyclized nanostructures are highly stable (Fig. 4b, c). The fraction of opened structures increased as the incubation temperature increased, which could be attributed to more peptide hydrolysis in the linker regions at high temperatures. We also examined the thermostability of Tri10t in bulk using SDS–PAGE and observed similar thermostability (Supplementary Fig. 8a, b). We then evaluated the stability of Tri10t in harsh chemical conditions. Tri10t was incubated in chaotropic (6 M guanidine hydrochloride at 70 °C for 1 h) and organic solvents (ethanol, acetone, and DMSO for 24 h) and then added directly to grids and fixed. STEM imaging showed that the nanostructures remained almost entirely intact in all of these solvents, demonstrating their stability in harsh chemical environments (Fig. 4c, d).

**Gold nanoparticle assembly using triangular nanostructures**. Finally, to demonstrate how these covalently assembled structures might be used as scaffolds for assembling other molecules of interest, we harnessed protein side-chain chemistry for site-specific labeling. Our designed Tri10t structure contains one cysteine residue located at each vertex of the triangular nanostructure, with no other cysteine residues on the edges. Tri10t was reacted with 1.4 nm, maleimide-functionalized gold nanoparticles (AuNPs) to form maleimide–thiol conjugates (Fig. 5a). After purification by gel filtration chromatography (Supplementary Fig. 9), the Tri10t–AuNP assembly was visualized by STEM using methylamine vanadate staining (Fig. 5b). As expected, the AuNPs

were site-specifically assembled at the three vertices of each triangle, proofing the concept of using these protein nanostructures as scaffolds. This method can be potentially extended to assemble AuNPs at other locations of a triangle by incorporating cysteine mutations into specific sites of the 3HB edges or into other geometries (e.g. square shape) by using nanostructures with alternative shapes.

**Discussion**

Herein we demonstrate the covalent assembly of individually folded 3HB proteins into hyperstable 2D protein nanostructures with triangle and square shapes using SI-mediated post-translational ligation. We tested different linker lengths for the triangle structures and found that shorter linkers lead to reduced structural flexibility, as expected, and that the linker length alone can explain the observed angle statistics with quantitative precision. The square structures provide an interesting symmetric trimodal angle distribution centered at 90° that is a reflection of the square and rhombic natures of these structures. In addition to the 3HB selected for this study, other individually folded proteins can be used as building blocks for covalent assemblies, greatly expanding the diversity and complexity of possible nanostructures. Natural discoveries and engineering efforts have produced a large library of orthogonal SIs[21] that can be used to assemble more complex structures. Furthermore, site-specific, post-translational, covalent assembly is not limited to only SIs; other biochemical tools like sortase[34] and SpyTag/SpyCatcher[35] can be used in place of or even together with SIs. Incorporating multiple orthogonal ligation domains for covalent assembly can lead to more complex geometries, including 3D shapes by ligation using multi-component junctions.

The exceptional stability and precise scaffolding capabilities of our covalently assembled protein nanostructures allow them to be used for valuable applications not possible for many other nanostructures, protein or not. Their stability in extreme thermal conditions makes them ideal candidates for use as scaffolds for assembling and spatially coordinating thermophilic enzymes that operate most efficiently in elevated temperatures, such as many cellulolytic enzymes[36]. These nanostructures could also be used to coordinate multi-step enzymatic reactions in harsh chemical conditions, acting as scaffolds for industrial enzymes in processes where organic solvents are unavoidable[37]. In addition to their thermal and chemical stability, these cyclized structures are also expected to be protected from proteolysis, especially by

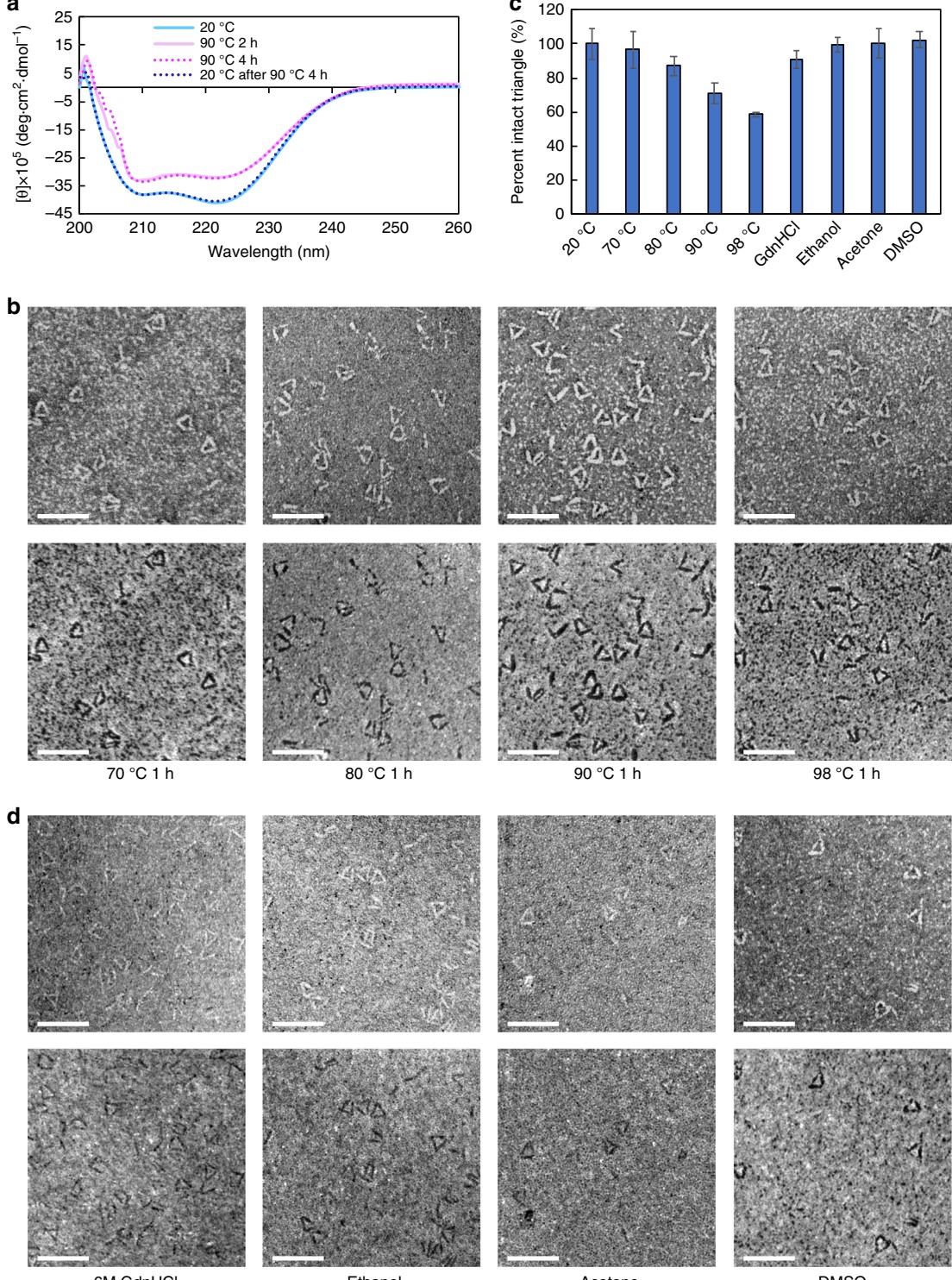

**Fig. 4** Thermal and chemical stability of the protein triangles. **a** CD spectra of the Tri10t sample incubated at 20 and 90 °C. **b** Bright-field (top) and dark-field (bottom) STEM images of protein triangles after 1 h incubation at different temperatures. Scale bars, 50 nm. **c** Percentages of 3HB copies contained in intact triangular nanostructures after incubation in different conditions, normalized by pre-incubation values. Error bars represent the s.d. ($n = 3$ denaturation experiment replicates). **d** Bright-field (top) and dark-field (bottom) STEM images of the protein triangles after incubation in chaotropic and organic solvents. Scale bars, 50 nm. Source data are provided as a Source Data file

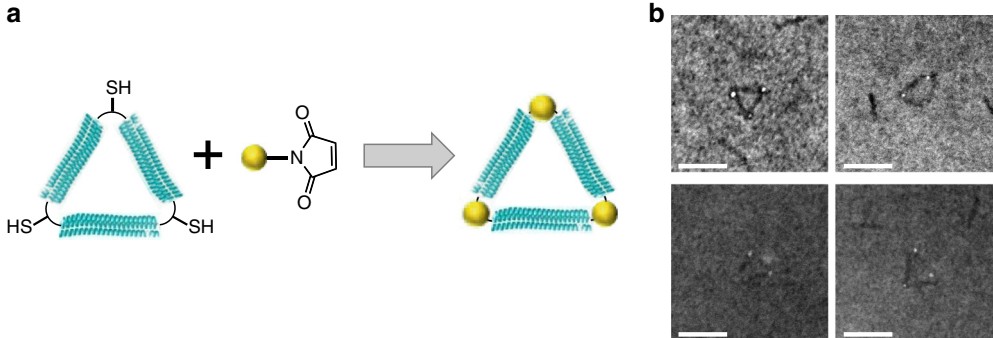

**Fig. 5** Assembly of gold nanoparticles (AuNPs) using Tri10t as a molecular scaffold. **a** Schematic depicting the conjugation of AuNPs to the vertices of the Tri10t protein nanostructures. **b** Dark-field STEM images showing AuNPs (lighter than background) attached to the three vertices of the protein nanostructures (darker than background). Scale bars, 20 nm

exopeptidases[19], making them prime candidates for biomedical applications such as the generation of advanced multivalent vaccine nanoparticles, which have been demonstrated to elicit improved immune responses[5,38].

## Methods

**Strains and growth conditions.** For molecular cloning, *Escherichia coli* NEB 10β was cultured at 37 °C in Luria-Bertani (LB) medium with 100 μg/mL ampicillin. For protein production, *E. coli* BL21(DE3) was cultured at 37 °C in Terrific Broth (TB) containing 24 g/L yeast extract, 20 g/L tryptone, 0.4% v/v glycerol, 17 mM $KH_2PO_4$, and 72 mM $K_2HPO_4$ with 100 μg/mL ampicilin.

**Chemicals and reagents.** Unless otherwise noted, all chemicals and reagents were obtained from Millipore Sigma. Plasmid purification and gel extraction kits were purchased from iNtRON Biotechnology. FastDigest restriction enzymes and T4 DNA ligase were purchased from Thermo Fisher Scientific.

**Plasmid construction.** Amino acid sequences of the orthogonal split inteins (SIs; Gp41-1, Cfa, and NrdJ) were obtained from published papers[22,23,39] and codon-optimized for expression in *E. coli* using the SI-Bricks method[40]. The optimized DNA sequences were chemically synthesized by Integrated DNA Technologies, Inc. (IDT) and PCR-amplified using different sets of primers (Supplementary Table 2 and Supplementary Dataset 1; also acquired from IDT) for Golden Gate DNA Assembly. The DNA sequence of the three-helix bundle (3HB) was amplified from the 3H5L_2 gene[24] using primers as listed (Supplementary Table 2 and Supplementary Dataset 1). PCR-amplified DNA fragments were inserted into expression plasmids using Golden Gate DNA Assembly[41]. The amino acid sequences of the resulting fusion proteins are listed in Supplementary Dataset 2. Sequences of all assembled plasmids were confirmed by DNA sequencing (Eurofins Genomics, USA).

**Expression and purification of 3HB-SI fusion proteins.** Sequenced plasmids were transformed into host cells for protein overexpression. Transformed cells were pre-cultured at 37 °C overnight in 5 mL TB containing 100 μg/mL ampicillin. Overnight cultures were used to inoculate 1 L TB with 100 μg/mL ampicillin. The cultures were grown at 37 °C until $OD_{600}$ reached 2 and were induced by the addition of 20% arabinose to a final concentration of 0.1%. Cells were then cultured at 18 °C for 20 h and harvested by centrifugation. To purify the 3HB-SI fusion proteins, cell pellets were resuspended in 1X PBS (pH 7.4) with 10 mM imidazole and lysed by sonication. Cell lysate was then centrifuged at 4 °C and 18,000×*g* for 30 min, and the precipitate was resuspended in Buffer A (1X PBS, 8 M urea, 10 mM Imidazole, 1 mM TCEP, pH 7.4) and loaded onto a pre-equilibrated Ni-NTA column (Qiagen). After loading, the column was washed with 10 column volumes (CV) of Buffer B (1X PBS, 8 M urea, 60 mM imidazole, 1 mM TCEP, pH 7.4), followed by washing with 10 CV of Buffer C (1X PBS, 8 M urea, 1 M NaCl, 1 mM TCEP, pH 7.4). The desired proteins were eluted with 10 CV of Buffer D (1X PBS, 8 M urea, 300 mM imidazole, 1 mM TCEP, pH 7.4), and the eluted fractions were dialyzed against 1X PBS.

**Synthesis and purification of protein triangles and squares.** To construct triangular nanostructures, two 3HB-SI building blocks (e.g. in the case of Tri18, $Cfa^C_{14}$-3HB-$NrdJ^N_{18}$ and $NrdJ^C_{18}$-3HB-$Gp^N_{18}$) were first mixed at a 1:1 ratio in Buffer E (1X PBS, 1 M NaCl, 1 mM TCEP, pH 7.4) at 37 °C for 8 h. The third building block (in this case $Gp^C_{18}$-3HB-$Cfa^N_{14}$) was then added in the same molar ratio and incubated at 37 °C for another 12 h to allow complete ligation. To

construct square nanostructures, two 3HB-SI building blocks ($Gp^C_{10}$-3HB-$Cfa^N_{10}$ and $Cfa^C_{10}$-3HB-$NrdJ^N_{10}$) were mixed at 1:1 ratio in Buffer E at 37 °C for 8 h. Meanwhile, the other two building blocks ($NrdJ^C_{10}$-3HB-$Cfa^N_{10}$ and $Cfa^C_{10}$-3HB-$Gp^N_{10}$) were also ligated using the same conditions. The two reaction mixtures were then mixed at a 1:1 molar ratio and incubated at 37 °C for another 12 h.

Because the histidine tag on each building block is removed during SI ligation, the cyclized protein nanostructures lacked a histidine tag and were thereby purified from both unreacted building blocks and partially reacted intermediates (e.g. dimers, trimers) by affinity chromatography. Each ligation product was loaded onto a pre-equilibrated Ni-NTA column, and the flow through containing the cyclized nanostructures was collected. The flow through was subsequently purified by SEC using a Superose 6 Increase 10/300 GL column (GE Healthcare). The samples were loaded with 1X PBS buffer with 500 mM NaCl and eluted with an isocratic elution.

**SDS-PAGE analysis.** All SDS–PAGE gels were 1 mm thick and discontinuous with a 6% stacking gel and 10% separating gel. Protein building block samples, ligation intermediate/side product, and circularized triangle and square were prepared in Laemmli sample buffer (2% SDS, 10% glycerol, 60 mM Tris, pH 6.8, 0.01% bromophenol blue, 50 μM DTT) and boiled at 100 °C for 15 min. The Dual Color Precision Plus Protein™ Standard (BioRad) was used as reference. Gels were run on Mini-PROTEAN Tetra Cells (Bio-Rad) in 1x Tris–glycine SDS buffer (25 mM Tris base, 250 mM glycine, 0.1% w/v SDS) until just before the dye front exited the gel. Gels were stained in Coomassie Blue solution (50% v/v methanol, 10% v/v acetic acid, 1 g/L Coomassie Brilliant Blue) for a minimum of 1 h at room temperature with gentle agitation and destained in destaining buffer (40% v/v methanol, 10% v/v acetic acid) for a minimum of 1 h. Gels were imaged on an Azure c600 Imager (Azure Biosystems). Uncropped and unprocessed scans of SDS–PAGE gels are included in the Source Data file.

**Negative stain electron microscopy.** A 10 μL drop of purified triangular or square nanostructures or of ligation intermediates/side products was applied to UV-treated (Novascan), pure carbon 400-mesh copper grids (Ted Pella, Inc.) and allowed to incubate for 5 min. Grids were then washed with three drops of Milli-Q water and a 10 μl drop of 0.75% uranyl formate, after which they were stained using a 10 μl drop of 0.75% uranyl formate, incubated for 3 min[42]. Liquids were wicked off of the grids between each step using filter paper, and the grids were allowed to air-dry. Samples were imaged on a STEM (JEM-2100F, JEOL, Japan). STEM images were collected using both a bright-field (BF) detector and a high-angle annular dark-field (HAADF) detector at 200 kV. Images were simultaneously recorded from both STEM detectors. The camera length was 12 cm and magnification was ×1,000,000.

**Data analysis by ImageJ.** For each sample, 100 triangles or squares were randomly selected by manually identifying triangle-shaped or square-shaped arrangements of 3HB proteins that had, based on the results from atomistic simulations, no more than 3.5 nm separation between each pair of adjacent 3HB edges. The interior angles for these cyclized structures were measured using ImageJ software, and the standard deviation of the three vertex angles within each individual triangle for the five triangular nanostructures was calculated as

$$SD = \sqrt{\frac{\Sigma(X_i - \langle 60 \rangle)^2}{n-1}}, \qquad (2)$$

where $n$ is 3 and $x_i$ is the vertex angle value of each individual triangle in that nanostructure. The standard deviation for all angles in triangular nanostructures

was calculated as

$$SD = \sqrt{\frac{\Sigma(X_i - 60)^2}{n-1}} \tag{3}$$

where $n$ is 300 and $x_i$ is the measured angle values.

**Atomistic simulations of protein linkers.** We performed atomistic simulations for a series of peptide linkers, modeled as autonomous units, and for the tethered systems conforming to the architecture of 3HB-linker-3HB. In all, we performed simulations for 14 different systems. In these simulations, the N- and C-termini of each of the peptides were capped using acetyl and N-methylamine groups, respectively. Details of the sequences used in each of the simulations are shown in Supplementary Table 5. The goal is to obtain converged, statistically robust descriptions of conformational ensembles for each of the simulated systems. Accordingly, we performed simulations using the ABSINTH implicit solvation model and forcefield paradigm[43] as implemented in the CAMPARI modeling suite v.2[44]. The ABSINTH model is interoperable with standard molecular mechanics forcefields and used parameters from the all atom OPLS-AA/L forcefield as implemented in the abs_opls3.2.prm parameter set within CAMPARI. For each peptide, we performed 200 independent Markov Chain Metropolis Monte Carlo (MC) simulations. The starting structures for these simulations were extracted randomly from a distribution of self-avoiding conformations. Each simulation consists of $5.1 \times 10^6$ MC steps, and the first $0.1 \times 10^6$ steps were discarded for equilibration. The atomic coordinates were registered every 5000 steps. The simulation temperature was set to 300 K, and the background NaCl concentration was set to ~150 mM while preserving charge neutrality. Ions are modeled explicitly while the solvent is treated in a mean field manner using the ABSINTH model. Our choice of solvation model, forcefield, and sampling protocols were motivated by the following considerations: We sought a combination of solvation model and forcefield with a proven track record of accuracy for simulating folded domains and disordered regions. Many of the established combinations of explicit solvent models and molecular mechanics models lack the accuracy to be applied to systems that include hyper-stable ordered domains and disordered regions with significant conformational heterogeneity. The ABSINTH model serves this purpose, as was shown recently in a high-throughput study of disordered regions tethered as linkers and/or tails to ordered domains[29]. Further, the computational demands increase with the size of the system, and, for molecules corresponding to the architecture 3HB-linker-3HB, simulations with explicit representations of solvent molecules become orders of magnitude more expensive when compared to simulations of the linkers alone. This increase in computational complexity is offset by the choice of the highly efficient ABSINTH model. Because the ABSINTH model relies on a fast estimation of conformational specific free energies of solvation, referenced as they are to experimentally derived free energies of solvation for model compounds, we are in a position to assess the interplay between solvation and desolvation effects through the interplay of the direct mean field interaction with the solvent and the polar term. Details of how the ABSINTH model works and why it works well have been previously published[28,43,45]. The attached key file that is interoperable with version 2.0 of CAMPARI enables reproduction of the simulation results.

To obtain the end-to-end distance distribution for each peptide, we computed the distances between the two $sp^3$ carbon atoms at the terminal acetyl and N-methylamide groups and constructed a histogram with bin size of 1 Å. The histogram was then converted into potential of mean force (PMF) by taking a logarithm and dividing it by $4\pi r^2$. We fitted the PMF with the fitting function,

$$f(x) = A + 3(x/R_0)^2, \tag{4}$$

to extract the Gaussian characteristic length $R_0$. If the fitting does not work well, it indicates that the chain cannot be assumed to be Gaussian, which was not the case (Supplementary Fig. 3). The secondary structure analysis was performed by the DSSP analysis module[46] implemented in the CAMPARI engine. The fractions of residues assigned as 'H' and 'E' were used as proxies to α and β contents, respectively.

**Coarse-grained simulations of protein nanostructures.** We developed a phenomenological, rods-and-springs coarse-grained model to perform simulations of triangular and square nanostructures. A schematic of the model for triangular nanostructures is shown in Supplementary Fig. 10. In the rod-and-springs representation, residues 6–238 of 3HB are modeled as rigid rods. These rods are of fixed length 114.32 Å, which corresponds to the distance between the two alpha carbons of residues 6 and 238 in the 3HB crystal structure. The rigid rod model is based on the assumption that the rod is not significantly distorted when compared to the flexible tethers, at least on experimentally relevant time scales. The linkers are modeled as Gaussian chains with an elastic energy written in terms of a harmonic potential:

$$E(x) = 2(x/R_0)^2; \tag{5}$$

note the factor of 2, which is appropriate given that the dimensionality is 2 for planar structures. Here, $R_0$ is determined by regression analysis where a second-order polynomial $f(x) = A - 3(x/R_0)^2$ is used to fit the PMF written in terms of the end-to-end distances extracted from atomistic simulations. The regression analysis

yields linker specific values for $R_0$ and these are used in the coarse-grained simulations.

Starting from an initial default structure, which is an equilateral triangle or square with side length of 114.32 Å, a MC procedure was used to equilibrate the system. At each MC step, one side is randomly picked among three and tilted by a random angle variable that follows a Gaussian distribution of mean 0 and standard deviation 0.1°. Then the tip-to-tip distance $x$ is converted into the Gaussian chain energy $E(x)$. The Gaussian chain energy is calculated for every linker, and the system energy is obtained by a simple sum of all 3 or 4 linker contributions. If the trial move gives lower energy, the program accepts the move. Otherwise, it accepts the move according to the Metropolis MC criterion. We performed 100 independent simulations for triangular structures, each of which consists of $10^7$ MC steps, and the coordinates were registered every $10^4$ steps, discarding the first $5 \times 10^5$ steps. For square structures, we also performed 100 simulations, but the step number was increased to $10^9$ and we discarded the first $10^8$ MC steps. The distributions of bending angles were constructed from the coordinate information, from which the standard deviation was also calculated.

**CD measurement.** CD measurements were performed on a JASCO J-810 CD spectrometer equipped with a Lauda RM 6 refrigerated circulator and a JASCO PTC-423S peltier. Each protein nanostructure was dissolved in 1X PBS at a concentration of 10 μM and loaded into a capped 1 mm quartz cuvette (Hellma, Germany). CD spectra were scanned from 200 to 260 nm using a 1 nm step, 1 nm bandwidth, 50 nm/min scanning speed, and 2 s response time. The stability of nanostructures was measured by scanning CD spectra from 200 to 260 nm at 20 or 90 °C after incubating for varying amounts of time as indicated. All CD spectra were recorded in triplicate and averaged.

**Thermal and chemical stability of protein nanostructures.** The triangular nanostructure Tri10t (5 μM in 1X PBS) was incubated for 1 h at four different temperatures (70, 80, 90, and 98 °C). Each sample was then mixed with Laemmli sample buffer and run in 1 mm SDS–PAGE gels in 1x Tris–glycine SDS buffer until just before the dye front exited the gel. Gels were then stained, destained, and imaged as before. Each sample was prepared and run on a gel three separate times. Protein band intensities were integrated using GelAnalyzer 2010 software (www. gelanalyzer.com). After incubation, samples were also immediately loaded to a UV-treated, pure carbon copper grid and prepared by 0.75% uranyl formate negative staining for STEM imaging. For each sample, ~150–400 copies of 3HB were randomly selected. The percentage of these 3HB copies that were contained in intact triangular nanostructures, as defined previously, was calculated. Percentages were reported after being normalized by the corresponding percentage calculated for an unincubated sample (prepared at 20 °C). The same experiment was repeated in triplicate for each sample to obtain mean percent intact triangle and standard deviation.

To evaluate the stability of these nanostructures in organic solvents, Tri10t was incubated at 50 nM in ethanol, acetone, or dimethylsulfoxide (DMSO) at 4 °C for 24 h. Samples were directly loaded to a carbon-coated copper grid and stained in triplicate as before, followed by STEM imaging. To assess the stability of these nanostructures in chaotropic conditions, Tri10t was dissolved at 500 nM in 1X PBS with 6 M guanidine hydrochloride (GdnHCl) and then incubated at 70 °C for 1 h. After incubation, the sample was diluted into 50 mM HEPES buffer (pH 7.5) and directly loaded to a carbon-coated copper grid and stained as before. All images were recorded by STEM and analyzed as described above. On each of the three girds per sample, ~50–120 copies of 3HB were counted for samples incubated in ethanol and GdnHCl, while ~30–50 copies were counted for samples incubated in acetone and DMSO due to protein aggregation. Percentages of 3HB copies contained in intact triangular nanostructures were calculated as before and normalized by the corresponding percentage from a sample incubated in 1X PBS with 500 mM NaCl. Samples were prepared and analyzed in triplicate.

**AuNP labeling of triangular nanostructure vertices.** AuNP labeling of triangular nanostructure vertices was performed based on the protocol provided for the Monomaleimido Nanogold® Labeling Reagent (Nanoprobe), with minor modifications. The Tri10t nanostructure purified using affinity chromatography was incubated in 1X PBS (pH 7.4) with 500 mM NaCl and 2 mM TCEP to keep the sulfhydryl group reduced at room temperature for 2 h. TCEP was then removed by SEC using a Superose 6 Increase 10/300 GL column and an isocratic elution buffer (0.02 M sodium phosphate, pH 6.5, 500 mM NaCl, and 1 mM EDTA). The reduced Tri10t sample (1 nmol) was then mixed with 0.2 ml Nanogold® reagent (6 nmol) in 0.02 M sodium phosphate (pH 6.5) with 500 mM NaCl. The reaction mixture was incubated at room temperature for 2 h. Tri10t nanostructure conjugates were separated from unreacted AuNPs by gel filtration chromatography using a Superose 6 Increase 10/300 GL column. The Tri10t-AuNP conjugates were eluted with 0.02 M sodium phosphate (pH 7.4) with 500 mM NaCl.

The purified Tri10t nanostructure conjugates were imaged by negative stain electron microscopy as described above, only using 2% methylamine vanadium (Nanoprobe) instead of 0.75% uranyl formate.

## Data availability
The source data underlying Figs. 2b, c, 3b, 4a, c, Supplementary Figs. 1–9, and Supplementary Table 3 are provided as a Source Data file. Other relevant data are available from the corresponding author upon reasonable request.

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

## Acknowledgements
We thank Prof. David Baker, Lauren Carter, Sydney Gordon, and Gustav Oberdorfer at the University of Washington for providing the 3HB plasmid and production advice. We thank Huafang Li at the Washington University in St. Louis (WUSTL) Institute of Materials Science and Engineering (IMSE); Remya Nair at the WUSTL Nano Research Facility (NRF); and James Fitzpatrick, Matthew Joens, Gregory Strout, and Daniel Geanon at the Washington University Center for Cellular Imaging (WUCCI) for help with sample preparation and TEM and STEM imaging. We thank Prof. Srikanth Singamaneni, Bin Dai, and Hongcheng Sun at WUSTL and Jonathan Galazka at NASA for useful suggestions on this work. We thank Prof. Matthew Lew for providing access to lab equipment for UV-treating the EM grids. STEM and TEM were performed within the facilities of IMSE, NRF, and WUCCI. This work was supported by the Office of Naval Research under the award number N000141512515 (to F.Z.), by a NASA Space Technology Research Fellowship under the award number 80NSSC18K1145 (to C.J.S. and F.Z.), the Human Frontier Science Program under the award number RGP0034/2017 (to R.V.P.), and the US National Science Foundation under the award number MCB-1614766 (to R.V.P.).

## Author contributions

F.Z. conceived the project. F.Z., W.B., and C.J.S. designed the experiments. W.B. prepared the nanostructure and performed the characterization. C.J.S. performed some STEM characterization. J.-M.C. and R.V.P. performed the computational simulations. All authors analyzed the data and wrote the paper.

## Additional information

**Competing interests:** The authors declare no competing interests.

