## [Peer Review File · Nature Communications]

Reviewers' comments:

Reviewer #1 (Remarks to the Author):

In this paper, the authors present an interesting study demonstrating the assembly of protein based building blocks via covalent ligation into 2D nanostructures of triangular and square shapes. This work is analogous to DNA origami which of course has a much larger body of past work than peptide/protein-origami. The authors claim that much of the past work in protein-origami is based on assembly via physical interactions like hydrogen bonding/electrostatics, and that such physically assembled nanostructures suffer from poor stability compared to the structures assembled via covalent bonds. Specifically in this work, the authors use split inteins¹⁸ (SI) to ligate multiple copies of a three-helix bundle (3HB)¹⁹ to create 2D triangular and square shaped structures. They place a proline-glycine dipeptide between 3HB and each of the SI groups of the fusion constructs to deter propagation of α -helical structure from the 3HB to the SI. They state that the α -helical structure of the 3HB forms a fairly rigid rod-like structure. They expect the flexible flanking regions of this rod-like helical domain to facilitate formation of the folded corners of the triangular or the square shaped nanostructure. To confirm the flexibility and lack of secondary structure formation of these linker regions, they use implicit solvent atomistic simulations. My first question to the authors is how they justify implicit solvent simulations? I understand these are meant to be fast simulations to sample quickly the conformations of the linkers, but those conformations should depend on the solvent quality. After all, polymer physics is all about quantifying scaling exponents of polymer conformations based on solvent quality. My next (related) question is about the choice of the force field. Was there a specific motivation for using the ABSINTH forcefield? I ask because it has been shown that the secondary structures of peptides can be sensitive to the choice of force field. I understand co-author Pappu and coworkers have done extensive work on this implicit solvation model and force field, but there is no justification in the method section for this choice, and the reader (especially one who may be less aware of this approach) may not understand the justification of this chosen forcefield and approach. In the method section the authors state "We performed 200 independent Markov Chain Metropolis Monte Carlo (MC) simulations using initial structures that were extracted randomly from a distribution of self-avoiding conformations." - The authors should expand this discussion a little as it is unclear how they create the initial structures. Their atomistic simulations results are all presented in the SI part of this manuscript. In the main paper they state "The results show that secondary structure is not frequently formed in the linkers (<10% of simulation time; Supplementary Table 2), and the linkers indeed behave like Gaussian random chains as their end-to-end distributions can be fitted well to a Gaussian model (Supplementary Fig. 3). This implies that the linker lengths are larger than the Kuhn length but smaller than the thermal blob size, meaning the chain behavior can therefore be simply explained by ideal-like interactions of Kuhn monomers." The validity of these results is closely linked to the chosen computational approach which has not been clearly justified/supported here.

The authors also use coarse-grained simulations to "understand the relationship between the structural regularity and the linker length". Again, the method section where these simulations are mentioned has very little detail; any reader with reasonable simulation background will not be able to replicate this work based on the few details mentioned. Coarse-grained is a very broad term and can be interpreted in so many different ways - an image/cartoon would have helped along with a little more detail in the text. For example, there is no way to guess what types of moves do the authors use before they calculate the energies for the metropolis Monte Carlo acceptance criterion. As for their results, they state "We tested various combinations of spring constants for the linker chains and obtained a linear relationship of the summation of characteristic lengths of three Gaussian-chain linkers (ΣR_0) to the standard deviation (σ) of angles (Supplementary Fig. 4): $\sigma = -2.37^\circ + (0.20^\circ/\text{\AA}) \times \Sigma R_0$. This implies that chain extensibility (as quantified by the characteristic length) almost additively contributes to the triangle flexibility (as quantified by the standard deviation)." I am not sure I find this surprising. Could the authors comment on why this was not obvious to them?

Overall, the topic of the paper is interesting and the choice of multiple length scale simulations is also commendable. The lack of justification for the chosen approaches (likely because these authors use them a lot) and the lack of details of their method makes it difficult to judge the

validity/correctness of their approach and the novelty of the work as a standalone piece.

Reviewer #2 (Remarks to the Author):

The authors have described a covalent methodology for the construction of protein origami nanostructures. While coiled-coil strategies have been the primary structural motifs for all protein based folding systems (Chem. Soc. Rev. 2018, 47, 3530), the covalent method proposed by the authors can be well-received and complementary to existing non-covalent type folding strategies. There are a few major concerns (listed below) which the authors should address the community before publication.

1. The authors effort to show a true representative microscopic image with defects is worth commendation. It is showed in the SEC and SDS-PAGE that the formation of the targeted nanostructures has a few side products. The authors should be encouraged to identify these, whether they are structural intermediates. Can this be improved/optimized? or is this a limitation? An open discussion would be highly beneficial for the audience and progress in this field.
2. The stability of the structures should be proven using bulk characterization techniques rather than imaging. This could be an SDS page (before and after) and the bands should be integrated to identify the % of nanostructures that are intact.
3. The covalent strategy is shown to work for triangle and square. A larger amount of shape profiles would be necessary to show the broad application of the technique. Currently, each junction seems to be limited to a bifunctional type conjugation. Can vertices of more junctions be created? i.e. tetrahedron with 3 junctions.

Reviewer #3 (Remarks to the Author):

In this paper, Bai and coworkers present a new method to form protein nanostructures from the bottom up using intein chemistry to link individual folded protein domains together covalently and site-specifically. The authors contrast their bottom-up approach to existing top-down strategies that utilize natural protein nanostructures, such as viral capsids, as templates and modify them through protein engineering. The authors claim that their method of producing nanostructures is advantageous because it may produce (a) fewer "incorrectly assembled or kinetically trapped intermediates" and (b) more thermostable structures as compared to non-covalently assembled protein nanostructures.

They specifically prepared and characterized two-dimensional triangular and square nanostructures constructed from a rod-like protein building block, namely a designed three-helix bundle (3HB) originally reported by the Baker laboratory. The rigid and thermostable 3HB protein was connected covalently through site-specific split-intein chemistry. To yield triangular or square shapes, three orthogonal split inteins were used. Triangle formation was performed in one step, whereas for the formation of squares, dimers were first generated and then subsequently linked together. In the triangle case, large standard deviations from the ideal vertex angle were found to correlate with the flexibility of the linker between the 3HB struts, a trend confirmed in coarse-grained simulations. For the squares, the linker that gave the most regular triangles afforded three major conformational populations, two rhombus-like structures consistent with the computationally predicted angles, and one unilateral square, which had not been predicted in the simulations. This inconsistency is not resolved in the paper but will "require further investigation." Finally, the authors show that the prepared nanostructures display high thermal stability "consistent with the CD spectral changes reported previously for 3HB."

The use of multiple orthogonal split inteins to assemble defined nanostructures is clever, and the effort that went into optimizing the linker to yield more regular shapes impressive. That said, the resulting structures are two-dimensional and it is not clear how these shapes will be immediately useful for the stated purposes, such as drug delivery, nanoparticle assembly scaffolds, or enzyme sensing. While the structures are aesthetically pleasing, the ultra-high stability simply mirrors the

properties as the starting 3HB protein from the Baker group. There seems to be no immediate benefit to the multimeric structures. If an application were shown, this aspect of the paper would be stronger.

Specific edits:

- Supplementary Figures 1 and 5: The depicted SDS-PAGE gels show significant impurities at lower molecular weights. Are these the dimer and/or trimer precursors originating from incomplete intein excision? The authors did not quantify these impurities, nor did they comment on their potential implications for characterization of the materials, for example for the thermostability measurements by CD.
- Line 43: reference 8 refers to Sasaki et al. (Nat. Commun. 2017). The designed capsids reported in this paper originate from bacterial microcompartments, not viruses.
- Line 44: the statement “the possible nanostructures created by these top-down approaches are strongly limited by the natural structures, making it difficult to construct novel or tailor-made structures” seems too general. Many such structures are amenable to directed evolution and have indeed produced novel, interesting structures.
- Line 55: Although it is true that protein nanostructures based on covalent linkages alone have not been widely explored, there are examples of protein nanostructures that make use of covalent domain linkages in conjunction with noncovalent binding interfaces, such as Padilla et al. (PNAS 2001), Shekhawat et al. (JACS 2009), Gradisar et al. (Nat. Chem. Biol. 2013), Susuki et al. (Nature 2016), Park et al. (Sci. Rep. 2017), Kobayashi et al. (ACS Synth. Biol. 2018), and others.
- Line 59: The assumption that “assembly using covalent bonds will generally yield more thermodynamically stable structures than assembly through non-covalent interactions” seems too general. This is dependent on many factors, such as the stability of the protein monomer, linker identity and chemistry, and others.
- Line 302: “100 triangles or squares were randomly selected”: more details on the methodology would be appropriate here. Were they selected manually or using an automated script? How were broken or incompletely formed structures classified?

Reviewers' comments:

Reviewer #1 (Remarks to the Author):

In this paper, the authors present an interesting study demonstrating the assembly of protein based building blocks via covalent ligation into 2D nanostructures of triangular and square shapes. This work is analogous to DNA origami which of course has a much larger body of past work than peptide/protein-origami. The authors claim that much of the past work in protein-origami is based on assembly via physical interactions like hydrogen bonding/electrostatics, and that such physically assembled nanostructures suffer from poor stability compared to the structures assembled via covalent bonds. Specifically in this work, the authors use split inteins¹⁸ (SI) to ligate multiple copies of a three-helix bundle (3HB)¹⁹ to create 2D triangular and square shaped structures. They place a proline-glycine dipeptide between 3HB and each of the SI groups of the fusion constructs to deter propagation of α -helical structure from the 3HB to the SIs. They state that the α -helical structure of the 3HB forms a fairly rigid rod-like structure. They expect the flexible flanking regions of this rod-like helical domain to facilitate formation of the folded corners of the triangular or the square shaped nanostructure. To confirm the flexibility and lack of secondary structure formation of these linker regions, they use implicit solvent atomistic simulations. My first question to the authors is how they justify implicit solvent simulations? I understand these are meant to be fast simulations to sample quickly the conformations of the linkers, but those conformations should depend on the solvent quality. After all, polymer physics is all about quantifying scaling exponents of polymer conformations based on solvent quality.

Answer: The ABSINTH implicit solvation model and forcefield paradigm has been discussed extensively in at least thirty separate publications. It has been used to model coil-to-globule transitions as a function of temperature for systems showing upper and lower critical solution temperatures. Overall, the model uses either temperature dependent or temperature independent free energies of solvation, derived from experimental data, to set the reference energy scales for model compounds that mimic functional groups within proteins. Changes to conformation alter the free energy of solvation, which in the fully solvated case would be a sum of the reference free energies of solvation. As conformations change, atomic specific solvation states are computed using solvent accessible volumes. These solvation states capture the effects of overlaps with solvation shells of multiple atoms around the atom of interest and is hence a many body description of chain solvation/desolvation. A polar term captures the effects of inhomogeneous desolvation effects. At its core, the conformation specific interplay between solvation and desolvation effects allows us to capture the effects of solvent quality as they change with temperature. If cosolutes or salts modulate changes to solvent quality, then these entities are modeled explicitly in ABSINTH. For a given temperature, the only way to modulate solvent quality in aqueous solvents is by changing sequence, and the effective quality of aqueous solvents for various sequences has been accurately predicted by ABSINTH – a statement that cannot be made for any of the explicit representations of water molecules (without bespoke parameterization) or implicit solvation models. A justification for the choice of ABSINTH has now been included in the revised methods. It is worth noting that a detailed treatment of why the choice was made and the virtues of ABSINTH over some other model is well beyond the scope of the current manuscript.

My next (related) question is about the choice of the force field. Was there a specific motivation for using the ABSINTH forcefield? I ask because it has been shown that the secondary structures of peptides can be sensitive to the choice of force field. I understand co-author Pappu and coworkers have done extensive work on this implicit solvation model and force field, but there is no justification in the method section for this choice, and the reader (especially one who may be less aware of this approach) may not understand the justification of this chosen forcefield and approach. In the method section the authors state "We performed 200 independent Markov Chain Metropolis Monte Carlo (MC) simulations using initial structures that were extracted randomly from a distribution of self-avoiding conformations." - The authors should expand this discussion a little as it is unclear how they create the initial structures. Their atomistic simulations results are all presented in the SI part of this manuscript. In the main paper they state " The results show that secondary structure is not frequently formed in the linkers (<10% of simulation time; Supplementary Table 2), and the linkers indeed behave like Gaussian random chains as their end-to-end distributions can be fitted well to a Gaussian model (Supplementary Fig. 3). This implies that the linker lengths are larger than the Kuhn length but smaller than the thermal blob size, meaning the chain behavior can therefore be simply explained by ideal-like interactions of Kuhn monomers." The validity of these results is closely linked to the chosen computational approach which has not been clearly justified/supported here.

Answer: ABSINTH has been extensively used to predict and reconstruct experimentally derived ensembles for disordered proteins and disordered regions within otherwise ordered proteins. A series of references to the relevant literature have now been included in the revised version. The linkers were chosen to be mostly disordered, given that ABSINTH is the optimal paradigm for quantifying secondary structure content (again this has been proven repeatedly in a series of publications, which we avoided citing due to space limitations and since this isn't a review article of ABSINTH). We have justified the choice of ABSINTH in a few short sentences within the main text. We direct the reviewer's attention to the CAMPARI documentation: "The possible degrees of freedom being randomized are the backbone dihedral angles of flexible chains and the rigid-body coordinates of the various molecules." We have also added relevant details to the subsection *Atomistic simulations of protein linkers* of the Methods section.

The authors also use coarse-grained simulations to "understand the relationship between the structural regularity and the linker length". Again, the method section where these simulations are mentioned has very little detail; any reader with reasonable simulation background will not be able to replicate this work based on the few details mentioned. Coarse-grained is a very broad term and can be interpreted in so many different ways - an image/cartoon would have helped along with a little more detail in the text. For example, there is no way to guess what types of moves do the authors use before they calculate the energies for the metropolis Monte Carlo acceptance criterion.

Answer: The only move type we employed was already explained in the Methods: "At each MC step, one side is randomly picked among three and tilted by a random angle variable that follows a Gaussian distribution of mean 0 and standard deviation 0.1°."

We have included a schematic that is now Supplementary Figure 10.

As for their results, they state "We tested various combinations of spring constants for the linker chains and obtained a linear relationship of the summation of characteristic lengths of three Gaussian-chain linkers (ΣR_0) to the standard deviation (σ)

of angles (Supplementary Fig. 4): $\sigma = -2.37^\circ + (0.20^\circ/\text{\AA}) \times \Sigma R_0$. This implies that chain extensibility (as quantified by the characteristic length) almost additively contributes to the triangle flexibility (as quantified by the standard deviation)." I am not sure I find this surprising. Could the authors comment on why this was not obvious to them?

Answer: Although it is expected that the two are positively correlated, the additive and linear relationship is not trivial, especially given that we are not dealing with free chains but constrained chains and the two variables are "lengths" and "angles," which are not usually interconvertible.

Overall, the topic of the paper is interesting and the choice of multiple length scale simulations is also commendable. The lack of justification for the chosen approaches (likely because these authors use them a lot) and the lack of details of their method makes it difficult to judge the validity/correctness of their approach and the novelty of the work as a standalone piece.

Reviewer #2 (Remarks to the Author):

The authors have described a covalent methodology for the construction of protein origami nanostructures. While coiled-coil strategies have been the primary structural motifs for all protein based folding systems (Chem. Soc. Rev. 2018, 47, 3530), the covalent method proposed by the authors can be well-received and complementary to existing non-covalent type folding strategies. There are a few major concerns (listed below) which the authors should address the community before publication.

1. The authors effort to show a true representative microscopic image with defects is worth commendation. It is showed in the SEC and SDS-PAGE that the formation of the targeted nanostructures has a few side products. The authors should be encouraged to identify these, whether they are structural intermediates. Can this be improved/optimized? or is this a limitation? An open discussion would be highly beneficial for the audience and progress in this field.

Answer: We thank the reviewer for the commendation of our efforts. We have performed additional experiments (SEC separation followed by SDS-PAGE and STEM analysis) to identify these impurities. In the revised manuscript, we demonstrated that the impurities were indeed monomeric, dimeric, and uncyclized trimeric 3HBs, as we originally suspected. Further discussion and data have been added to the manuscript (main text lines 116-132 and Supplementary Fig 2). In brief, we believe that the impurities are composed of 1) side products whose Int^C and/or Int^N groups have been cleaved off prior to normal SI ligation, which is in agreement with previous reports¹, and/or 2) unreacted monomer and intermediates, which can be explained by previously reported ligation yields of the SIs used (85-95%)².

If necessary for future studies, side reactions could be potentially reduced by identifying alternative SI groups with reduced side reaction rates or by optimizing the intein and/or extein residues of SIs, as has been done to improve other characteristics of SI ligation, like ligation kinetics and SI thermodynamic stability³.

References:

1. Shi J, Muir TW. Development of a tandem protein trans-splicing system based on native and engineered split inteins. *J Am Chem Soc* **127**, 6198-6206 (2005).
2. Shah NH, Muir TW. Inteins: nature's gift to protein chemists. *Chem Sci* **5**, 446-461 (2014).
3. Stevens AJ, Sekar G, Shah NH, Mostafavi AZ, Cowburn D, Muir TW. A promiscuous split intein with expanded protein engineering applications. *Proc Natl Acad Sci U S A* **114**, 8538-8543 (2017).

2. The stability of the structures should be proven using bulk characterization techniques rather than imaging. This could be an SDS page (before and after) and the bands should be integrated to identify the % of nanostructures that are intact.

Answer: As the reviewer requested, we have performed additional SDS-PAGE analysis and added the results from densitometric analysis to the revised manuscript. Please see lines 220-222 and Supplementary Fig. 8 for detail.

3. The covalent strategy is shown to work for triangle and square. A larger amount of shape profiles would be necessary to show the broad application of the technique. Currently, each junction seems to be limited to a bifunctional type conjugation. Can vertices of more junctions be created? i.e. tetrahedron with 3 junctions.

Answer: Although SIs are inherently bifunctional, other junction types with more than two functions can be created by integrating our bifunctional SIs with other biochemical tools like sortase and SpyTag/SpyCatcher, thereby increasing the dimension of covalently-assembled protein nanostructures from 2D to 3D. More discussion relating to this point has been added to the manuscript. Please see line 254-256 for details. Exploring higher dimensional structures requires additional layers of design considerations and is beyond the scope of this study, which focuses on the initial proof-of-concept for covalently-linked protein nanostructures and developing the design rules for controlling flexibility of 2D structures.

Reviewer #3 (Remarks to the Author):

In this paper, Bai and coworkers present a new method to form protein nanostructures from the bottom up using intein chemistry to link individual folded protein domains together covalently and site-specifically. The authors contrast their bottom-up approach to existing top-down strategies that utilize natural protein nanostructures, such as viral capsids, as templates and modify them through protein engineering. The authors claim that their method of producing nanostructures is advantageous because it may produce (a) fewer "incorrectly assembled or kinetically trapped intermediates" and (b) more thermostable structures as compared to non-covalently assembled protein nanostructures.

They specifically prepared and characterized two-dimensional triangular and square nanostructures constructed from a rod-like protein building block, namely a designed three-helix bundle (3HB) originally reported by the Baker laboratory. The rigid and thermostable 3HB protein was connected covalently through site-specific split-intein chemistry. To yield triangular or square shapes, three orthogonal split inteins were used. Triangle formation was performed in one step, whereas for the formation of squares,

dimers were first generated and then subsequently linked together. In the triangle case, large standard deviations from the ideal vertex angle were found to correlate with the flexibility of the linker between the 3HB struts, a trend confirmed in coarse-grained simulations. For the squares, the linker that gave the most regular triangles afforded three major conformational populations, two rhombus-like structures consistent with the computationally predicted angles, and one unilateral square, which had not been predicted in the simulations. This inconsistency is not resolved in the paper but will “require further investigation.” Finally, the authors show that the prepared nanostructures display high thermal stability “consistent with the CD spectral changes reported previously for 3HB.”

The use of multiple orthogonal split inteins to assemble defined nanostructures is clever, and the effort that went into optimizing the linker to yield more regular shapes impressive. That said, the resulting structures are two-dimensional and it is not clear how these shapes will be immediately useful for the stated purposes, such as drug delivery, nanoparticle assembly scaffolds, or enzyme sensing. While the structures are aesthetically pleasing, the ultra-high stability simply mirrors the properties as the starting 3HB protein from the Baker group. There seems to be no immediate benefit to the multimeric structures. If an application were shown, this aspect of the paper would be stronger.

Answer: We thank the reviewer for the critical evaluation of our work. To address the reviewer’s concern, we performed additional experiments to demonstrate the application of our protein nanostructures as scaffolds for the site-specific assembly of nanoparticles. Using cysteine side-chain reactions, 1.4 nm maleimide-functionalized gold nanoparticles were specifically assembled at the three vertices of our triangular shapes. Please see lines 227-240 and Fig 5 for details. We believe that our method can be readily extended to assembling other molecules, such as other inorganic nanoparticles, enzymes, epitopes, motor proteins, etc. with defined geometry and nanometer resolution for various applications.

Specific edits:

- Supplementary Figures 1 and 5: The depicted SDS-PAGE gels show significant impurities at lower molecular weights. Are these the dimer and/or trimer precursors originating from incomplete intein excision? The authors did not quantify these impurities, nor did they comment on their potential implications for characterization of the materials, for example for the thermostability measurements by CD.

Answer: To clearly answer the reviewer’s question, we have performed additional experiments (SEC separation followed by SDS-PAGE and STEM analysis) to identify these impurities. In the revised manuscript, we demonstrated that the impurities were indeed monomeric, dimeric, and uncyclized trimeric 3HBs, as we originally suspected. Further discussion and data have been added to the manuscript (main text lines 116-132 and Supplementary Fig 2). In brief, we believe that the impurities are composed of 1) side products whose Int^C and/or Int^N groups have been cleaved off prior to normal SI ligation, which is in agreement with previous reports¹, and/or 2) unreacted monomer and intermediates, which can be explained by previously reported ligation yields of the SIs used (85-95%)².

After purification, concentrations of these low molecular weight species were low compared to that of the purified nanostructures. We do not think these impurities would significantly affect our

characterization of the protein nanostructures. For thermostability measurements, the 3HB has previously been demonstrated using CD to be stable at 95°C in aqueous conditions and in up to 7 M GdnHCl at 80°C⁴. We therefore do not expect the impurities to impact our CD measurements. Additionally, the small amount of impurities was accounted for during STEM-based stability analysis; all our reported results were first normalized by the percentage of intact triangular nanostructures in untreated samples (prepared at 20°C). Please see lines 208-222 for more details.

Reference

1. Shi J, Muir TW. Development of a tandem protein trans-splicing system based on native and engineered split inteins. *J Am Chem Soc* **127**, 6198-6206 (2005).
2. Shah NH, Muir TW. Inteins: nature's gift to protein chemists. *Chem Sci* **5**, 446-461 (2014).
4. Huang PS, Oberdorfer G, Xu C, Pei XY, Nannenga BL, Rogers JM, DiMaio F, Gonen T, Luisi B, Baker D. High thermodynamic stability of parametrically designed helical bundles. *Science* **346**, 481-485 (2014).

- Line 43: reference 8 refers to Sasaki et al. (Nat. Commun. 2017). The designed capsids reported in this paper originate from bacterial microcompartments, not viruses.

Answer: We have modified the manuscript text to better align with the selected reference.

- Line 44: the statement “the possible nanostructures created by these top-down approaches are strongly limited by the natural structures, making it difficult to construct novel or tailor-made structures” seems too general. Many such structures are amenable to directed evolution and have indeed produced novel, interesting structures.

Answer: While we recognize that many studies have demonstrated ways to produce novel structures by altering natural structures, we affirm that constructing truly tailor-made structures by modifying natural structures still presents major challenges. We have reworded this sentence as follow:

“However, because the extent of changes that can be made to these natural structures is fairly limited, rationally designing tailor-made structures through top-down approaches remains difficult.”

- Line 55: Although it is true that protein nanostructures based on covalent linkages alone have not been widely explored, there are examples of protein nanostructures that make use of covalent domain linkages in conjunction with noncovalent binding interfaces, such as Padilla et al. (PNAS 2001), Shekhawat et al. (JACS 2009), Gradisar et al. (Nat. Chem. Biol. 2013), Susuki et al. (Nature 2016), Park et al. (Sci. Rep. 2017), Kobayashi et al. (ACS Synth. Biol. 2018), and others.

Answer: We agree with the reviewer and have added more discussion regarding this point to the manuscript, along with these references.

- Line 59: The assumption that “assembly using covalent bonds will generally yield more thermodynamically stable structures than assembly through non-covalent interactions” seems too general. This is dependent on many factors, such as the stability of the protein monomer, linker identity and chemistry, and others.

Answer: We thank the reviewer for pointing out a portion of misleading text in our manuscript. We agree that the stability of assembly interfaces is only one of the many determinants of the overall stability of the structures and have reworded the manuscript to more clearly and accurately describe the benefits of covalent assembly:

“Furthermore, it stands to reason that structures assembled using covalent bonds will generally have higher thermodynamic stability at the assembly interfaces than structures assembled through non-covalent interactions, thus enabling broader applications in extreme temperatures and chemical conditions.”

- Line 302: “100 triangles or squares were randomly selected”: more details on the methodology would be appropriate here. Were they selected manually or using an automated script? How were broken or incompletely formed structures classified?

Answer: We have provided further details about our method in the manuscript, which now reads:

“For each sample, 100 triangles or squares were randomly selected by manually identifying triangle- or square-shaped arrangements of 3HB proteins that had, based on the results from atomistic simulations, no more than 3.5 nm separation between each pair of adjacent 3HB edges. The interior angles for these cyclized structures were measured using ImageJ software.”

REVIEWERS' COMMENTS:

Reviewer #1 (Remarks to the Author):

The authors have addressed all of my comments/questions in a satisfactory manner. I am okay with the paper being accepted for publication.

Reviewer #2 (Remarks to the Author):

The authors have sufficiently addressed my concerns and the manuscript can be published as it is.

Reviewer #3 (Remarks to the Author):

The authors have appropriately addressed the concerns and suggestions raised in my original review. The assembly of gold nanoparticles on a triangular scaffold, in particular, is a good addition. I am therefore happy to recommend the revised manuscript for publication in Nature Communications.